# Different Oral Appliance Designs Demonstrate Different Rates of Efficacy for the Treatment of Obstructive Sleep Apnea: A Review Article

**DOI:** 10.3390/bioengineering12020210

**Published:** 2025-02-19

**Authors:** Leonard A. Liptak, Edward Sall, Sung Kim, Erin Mosca, Shouresh Charkhandeh, John E. Remmers

**Affiliations:** ProSomnus Sleep Technologies, Pleasanton, CA 94588, USA

**Keywords:** Obstructive Sleep Apnea, Oral Appliance Therapy, Respiratory Medicine

## Abstract

Obstructive Sleep Apnea afflicts an estimated 1 billion people worldwide. Untreated, Obstructive Sleep Apnea is linked with elevated levels of mortality, decreased quality of life and increased economic costs. However, several large population studies demonstrate that the efficacy of Continuous Positive Airway Pressure therapy, the most frequently prescribed treatment for Obstructive Sleep Apnea, is compromised by frequent refusals and terminations. As a result, healthcare providers are evaluating non-CPAP treatment options. Oral Appliance Therapy has emerged as a leading non-CPAP treatment for patients with Obstructive Sleep Apnea. Historically, healthcare providers have questioned the efficacy of Oral Appliance Therapy. Dozens of Oral Appliances are available to healthcare providers, with many contemporary Oral Appliances featuring improved designs, materials and technologies. This review investigates whether Oral Appliance design matters; do different Oral Appliance designs demonstrate different rates of efficacy? To the best of the authors’ knowledge, this is the first review to exclusively focus on scientific papers that report treatment success with Oral Appliances as a residual Apnea Hypopnea Index of less than 10 events per hour. Out of 272 source papers, the 27 papers included in this review encompass a pooled sample of 3799 patients treated with six distinctly different categories of Oral Appliance designs. Chi-squared and two-sided Fisher’s exact tests indicate significant differences in efficacy amongst Oral Appliance designs. These findings suggest that certain Oral Appliance designs can enable highly efficacious treatment for patients with Obstructive Sleep Apnea. Given these findings, healthcare providers should consider design when selecting an oral device for patients diagnosed with Obstructive Sleep Apnea.

## 1. Introduction and Background

Obstructive Sleep Apnea (“OSA”) is a highly prevalent, chronic, respiratory disease, afflicting an estimated 1 billion people worldwide [1]. Untreated, OSA is associated with elevated levels of morbidity, mortality, economic expenses [2] and reductions in quality of life [3]. Many healthcare providers are revisiting Oral Appliance Therapy (“OAT” also called “Mandibular Advancement” or “Mandibular Repositioning”) as a treatment modality for OSA, largely motivated by clinically and scientifically demonstrated limitations with Continuous Positive Airway Pressure (“CPAP”) treatment, and encouraging results with contemporary Oral Appliances (“OA”) that feature improved designs, technologies and materials. Studies associate CPAP with low compliance rates [4,5] and negative heart health outcomes for certain types of patients [6]. Several brands of CPAP devices have been recalled by the US Food and Drug Administration [7]. However, some healthcare providers remain skeptical of non-CPAP treatments and continue prescribing CPAP more frequently than non-CPAP treatment options [8].

This is the first review, to the best of the authors’ knowledge, to evaluate the association between different OA device designs and treatment efficacy. In this review, efficacy is defined as the percentage of patients who achieve a residual AHI of less than 10 events per hour.

Prescription of OAT is a standard recommendation for the treatment of OSA for adult patients who are intolerant of CPAP or prefer alternative treatment [9]. Research establishes that OAT is non-inferior to CPAP in terms of cardiovascular and neurobehavioral outcomes [10], systolic blood pressure [9], diastolic blood pressure [9], mean arterial blood pressure [11] and overall health outcomes [12].

OAT is covered by most commercial medical insurance plans in the United States [13] and government-sponsored programs in several countries around the world, including Medicare in the USA [14]. Successful OAT response is also associated with improved cardiac autonomic adaptability in patients with OSA [15], improved cognition [16] and significant reverse left ventricular hypertrophic remodeling [17].

It is thought that the greater Apnea Hypopnea Index (“AHI”) events per hour reduction in CPAP is offset by the superior adherence to OAT, resulting in similar, non-inferior, mean disease alleviation [18]. It is hypothesized that improving adherence is necessary to improve the therapeutic effectiveness of CPAP. Conversely, it is hypothesized that improving efficacy is necessary to improve the therapeutic effectiveness of OAT [19], hence the significance of evaluating OA design as a determinant of OAT efficacy.

There are at least two research streams on the topic of improving the efficacy of OAT for the treatment of OSA. The first stream involves patient selection, preselecting patients based on endotypical and phenotypical characteristics [20].

The second stream involves the impact of OA design on the physiologic mechanisms of action associated with using OAT to prevent airway collapse. Practice guidelines establish that successful OAs are those that 1. guide, 2. stabilize and 3. titrate the mandible into a therapeutic position that reduces the risk of airway collapse [21]. Two studies conclude that OA designs that guide the mouth closed and minimize the separation of the maxillary and mandibular teeth are linked with higher levels of efficacy [22,23]. Others associate higher levels of efficacy with OA designs that guide the tongue into a beneficial anterior position [24,25,26]. The implication is that OA designs that embody these characteristics are more likely to be efficacious than those that do not.

It should be noted that other reviews find no statistically significant differences in efficacy based on OA device designs [27]; however, they acknowledge clinically meaningful differences related to OA design.

## 2. Oral Appliance Design Categories

This review focuses on custom-made, titratable OA designs. OAs can be custom-made, or prefabricated. Prefabricated OAs are those that are mass pre-manufactured without a specific patient in mind. Custom-made OAs are made individually for a specific patient, based on the unique dental and airway anatomy of the patient. Clinical practice standards and guidelines recommend the prescription of custom-made titratable devices over prefabricated OAs [9].

For this review, custom-made, titratable, OA designs were organized into six categories according to how each device is engineered to achieve mandibular repositioning, the cardinal mechanism associated with achieving efficacious outcomes. The six categories of OA designs are: 1. Lateral Push, 2. Lateral Pull, 3. Interlocking Dorsal, 4. Precision Post, 5. Mono/Twin Block and 6. Anterior Pull.

Figure 1 provides illustrations for each of these six categories of OAT designs.

### 2.1. Lateral Push

Lateral Push OA designs have metal, piston-like, rod mechanisms, known as Herbst arms, on each side of maxillary and mandibular overlays that sit on the maxillary and mandibular dentitions, respectively. Lateral Push OAs reposition the mandible by pushing the mandibular overlay component forward, using the maxillary overlay as an anchor. Titration of the mandibular position is achieved by turning displacement screws embedded in the metal Herbst rods.

### 2.2. Lateral Pull

Lateral Pull OA designs have mechanisms, typically elastomeric or nylon straps, on each side of the maxillary and mandibular overlays that sit on the maxillary and mandibular dentitions, respectively. Lateral Pull OAs reposition the mandible by pulling the mandibular overlay component forward, using the maxillary overlay as an anchor. Titration of the mandibular position is achieved by swapping out/in elastomeric or nylon straps of different lengths.

### 2.3. Interlocking Dorsal

Interlocking Dorsal OA designs have angled dorsal posts that extend occlusally from each side of the mandibular overlays. These dorsal posts reposition and stabilize the mandible by interlocking with block features affixed to either side of the maxillary overlay. Titration of the mandibular position is achieved by turning displacement screws embedded into the block mechanisms (e.g., 10 turns of the screw yield +1 mm of mandibular repositioning).

### 2.4. Precision Post

Precision Post OA designs feature dual posts that extend occlusally from each side of the maxillary and mandibular overlays that sit on the maxillary and mandibular dentitions, respectively. The dual posts interlock perpendicularly to precisely guide, reposition and stabilize the mandible at the prescribed location to minimize airway collapse. Titration of the mandible is achieved iteratively by swapping out a maxillary or mandibular overlay for a new overlay that contains a different mandibular repositioning setting (e.g., +1 mm of mandibular repositioning).

### 2.5. Mono/Twin Block

Mono/Twin Block OA designs reposition and stabilize the mandible with maxillary and mandibular overlays that are fused together (Mono-Block) at the occlusal surface of each overlay, or mated (Twin Block) by interlocking block components affixed to the occlusal surfaces of the maxillary and mandibular overlays. Titration of the mandibular position is achieved by swapping out an overlay for new overlays that contain different settings.

### 2.6. Anterior Pull

Anterior Pull OA designs use maxillary and mandibular overlay components that sit on the maxillary and mandibular dentitions, respectively, and feature an anterior fixation mechanism, typically a screw or a nylon strap, that repositions the mandible using the opposing overlay as an anchor. Titration of the mandibular position is achieved by turning the anterior displacement screw or by swapping out the strap for a new strap with a different length.

## 3. Review

### 3.1. Objective

The objective of this review article is to interrogate our thesis that different OA designs are linked with significantly different rates of efficacy. This thesis is based upon the findings of several prior comparative clinical studies linking improved levels of efficacy with specific OA design characteristics, including promoting mouth closing, minimizing teeth separation, precise mandibular repositioning and stabilization, and anterior tongue posturing [21,22,23,24,25,26].

### 3.2. Search Strategy

A literature search was conducted, drawing upon the references identified in four systemic reviews on OAT for the treatment of OSA. The four “input” references were Clinical Practice Guideline for the Treatment of Obstructive Sleep Apnea and Snoring with Oral Appliance Therapy [9], Definition of an Effective Oral Appliance [20], Mandibular Advancement Device: A Systemic Review on Outcomes [28] and Precision vs. Traditional Oral Appliance Therapy: A Comparison of Efficacy [29]. This search strategy of drawing upon articles contained in existing reviews was selected as a safeguard against selection bias. Each of these reviews had previously identified articles relevant to OA efficacy and applied established screening criteria including the GRADE system, Cochrane Risk Bias, Network–Ottawa Scale, and Rand/UCLA Appropriateness Method. Figure 2 below illustrates the search strategy and workflow.

### 3.3. Selection Criteria

The following selection criteria were applied to the universe of source articles.

To be selected, the article must meet the following criteria:Primarily evaluate the efficacy of Oral Appliance Therapy devices;Describe the interventional OAT device in sufficient detail to definitively categorize the Oral Appliance design;Define and report therapeutic success as the percentage of patients achieving a residual AHI of less than 10 events per hour.

### 3.4. Definition of Therapeutic Efficacy

Defining the success of the treatment of OSA with OAT is not a straightforward endeavor [30]. Several articles define efficacy as a residual AHI value of less than five events per hour, which is typically considered a CPAP definition of success. Others use Sher’s criteria for surgical success, which calls for a residual AHI of less than 20 events per hour and a 50% improvement relative to baseline [31]. Others report the mean percent reduction in AHI events per hour for the study population as a definition of success. Each of these definitions have merits and limitations.

A residual AHI of less than 10 events per hour is used in this review. An AHI of less than 10 events per hour is a polysomnographic metric commonly used to assess the efficacy of OAT [32].

It should be noted that recent scientific investigations have scrutinized the utility of the AHI as a definition for success. Investigations have concluded that the AHI is a poor surrogate for predicting health outcomes [33]. Nonetheless, the AHI ris the most widely utilized metric diagnosing OSA, determining insurance coverage, and managing treatment success.

### 3.5. Collection of Data

Data related to the sample size, the percentage of patients who achieved a residual AHI of less than 10 events per hour, the design of the interventional OAT device and other notable study features were collected, organized and summarized in a database. Best efforts were made to collect data on mandibular starting location, titration schedule, sleep testing approaches, OSA severity and other metrics that are thought to be important determinants of OA efficacy. However, data for these variables were sporadically included in the source papers, limiting the ability to conduct post hoc analyses.

### 3.6. Statistical Methods

Chi-squared and two-sided Fisher’s exact tests were selected to evaluate the significance of differences between the six categories of device designs. The *p*-value threshold for the statistical significance of the two-sided Fisher’s exact tests was adjusted to 0.00333 to account for the multiple comparisons (0.05/15 comparisons).

### 3.7. Search Strategy Results

The bibliographies of the four systemic reviews encompassed 272 articles. Eliminating duplicates yielded 205 unique articles. A total of 27 articles remained for inclusion in the review after eliminating articles that did not report efficacy according to the AHI < 10 criteria. See Figure 2 for the search strategy flowchart used for this review.

The article with the largest sample size had 601 subjects. The article with the smallest sample size had 16 subjects. Four articles with a total of 113 subjects reported the efficacy of Lateral Push OAT devices. Six articles encompassing 765 subjects reported the efficacy of Lateral Push devices. Six articles encompassing 366 subjects reported the efficacy of Interlocking Flange devices. Nine articles involving 729 subjects reported the efficacy of Precision Post devices. Seven articles involving 723 subjects reported on Twin/Mono-Block devices. Five articles involving 938 subjects reported efficacy for Anterior Pull devices. See Table 1 for a complete list of reference articles.

### 3.8. Statistical Analysis

A chi-squared test revealed statistically significant differences (*p* < 0.000001) between the Dual Post OA design and the other OA device design categories.

The two-sided Fisher’s exact tests indicated significant differences in efficacy for the Precision Post OA category relative to the other five OA design categories: Lateral Push, Lateral Pull, Interlocking Dorsal, Mono-Twin Block and Anterior Pull.

The two-sided Fisher’s exact tests also indicated significant differences in efficacy for the Anterior Pull OA design relative to Lateral Push (*p* = 0.000441), Lateral Pull (*p* = 0.000119) and the Mono/Twin Block (*p* = 0.000401) OA design categories.

The two-sided Fisher’s exact test did not indicate significant differences in efficacy amongst any of the other device design categories. Table 2 contains a summary of Fisher’s exact test results for statistical significance.

## 4. Descriptive Statistical Results

Across all device categories, 63% of the 3768 subjects included in this review of 27 articles achieved a residual AHI of less than 10 events per hour. The median efficacy value was 65% relative to a performance threshold of AHI < 10 events per hour. The maximum efficacy reported by any individual article was 95% of patients achieving an AHI < 10. The minimum efficacy reported by any individual article was 30% achieving an AHI < 10. The overall standard deviation was 0.17 across all included papers.

## 5. Efficacy by Device Design Category

As illustrated in Figure 3, the Precision Post device design category was associated with the greatest median efficacy, with 87% of the 729 subjects achieving a residual AHI of less than 10 events per hour. The results were statistically significant (*p* < 0.000001) relative to the other OA designs. The Precision Post OA design also demonstrated the lowest standard deviation of 0.04. Conversely, the Lateral Push device design category was associated with the lowest median efficacy, with 55% of 113 subjects achieving a residual AHI of less than 10 events per hour. Lateral Pull, Anterior Pull, Mono/Twin Block and Interlocking Dorsal device design categories were associated with achieving 56%, 56%, 59% and 63% median residual AHI of less than 10 events per hour, respectively. See Figure 3 for an illustration of efficacy by OA design category.

## 6. Subjective Device Design Observations

What factors might explain the statistically significant differences between the Precision Post OA designs and the other OA designs? The Precision Post OA design seems better suited to perform the mechanisms of action that prior research associates with higher levels of efficacy. Specifically, the Precision Post design seems better engineered to stabilize the mandible in the prescribed therapeutic position throughout the night, encourage mouth closure and minimize teeth separation.

Several categories of OA designs seem less equipped to perform the mechanisms of action linked with higher levels of efficacy. For example, the Lateral Push, the Lateral Pull and Interlocking Dorsal OA designs seem less capable of stabilizing the mandible should the patient open their mouth during sleep. Such OA design limitations might also result in the mandible rotating into a less therapeutic position, allowing the mouth to open and the teeth to separate, thereby reducing the dilation of the airway.

Although not obvious from the line drawings in Figure 1, another potential explanation for the difference in efficacy is that the Precision Post OA design has less material on the lingual side of the dental overlays than the other designs, allowing more room for a beneficial anterior tongue position, a variable that is linked to greater levels of OA efficacy [24]. Many OA designs have significant material on the lingual side of the patient’s teeth, limiting space for the tongue and potentially discouraging the beneficial anterior tongue orientation that is associated with higher levels of efficacy.

Differences in OA materials are another possible explanation. Venema et al. report that OAs made from thermoplastic materials are associated with reduced levels of treatment success [28]. However, it is unclear if Venema was referring to “thermoplastic” as a material characteristic or “thermoplastic” as a generic description of non-custom OAs. Materials were not specifically considered in the OA design categories for this review due to lack of available data, though OA devices included in this review were certainly made from a range of materials, including polyamide-12 nylon, dual laminates, polymethylmethacrylate and medical-grade class VI polymers [51,62].

### Limitations

There are several limitations in this review article. One limitation involves the lack of data necessary to conduct post hoc analyses for variables that could significantly impact efficacy values. For example, it is thought that baseline OSA severity, starting mandibular positions and sleep testing thresholds can have significant impacts on efficacy values. However, only a handful of the source documents for this systemic review documented data for these key variables. This limited the ability to conduct post hoc analyses to understand the significance of these variables with respect to OA efficacy.

Another limitation, which is somewhat endemic for review articles, is the general variability in study methods. These articles contain protocol variabilities or are silent on clinical protocols with respect to baseline testing, follow-up testing, mandibular positioning protocols, device selection rationale, device titration protocols, patient anthropomorphics, disease severity profiles and more. These differences could translate into biases. However, as previously mentioned, the sporadic disclosure of these variables across the source papers impaired the ability to control for these factors in post hoc analyses.

The linkage between device design differences and efficacy being implied and not directly tested is another limitation to consider. Several of the source papers do directly control for OA design differences. Comparing a sham device that embodies the OA design features linked with higher OA efficacy against a sham device that does not could be an informative follow-up study.

Bias is another potential study limitation. The authors are involved in OAT and OA devices. The search strategy for this review was selected to mitigate the risk of bias and enable reproducibility. However, a similar review from independent sources would be a welcome follow-up.

## 7. Conclusions

OA design matters for the treatment of OSA with OAT. This review finds that specific OA designs are associated with significantly different levels of treatment efficacy, with design elements that correlate with the OA design characteristics identified in prior controlled clinical studies. Certain OA designs seem capable of providing a highly effective non-CPAP treatment option for patients diagnosed with OSA.

Healthcare providers should consider OA design when selecting, prescribing, delivering and managing patients with OSA. This review also suggests that relevant medical guidelines should go beyond the basic distinction between custom and non-custom OAT, and titratable and non-titratable device designs.

## Figures and Tables

**Figure 1 bioengineering-12-00210-f001:**
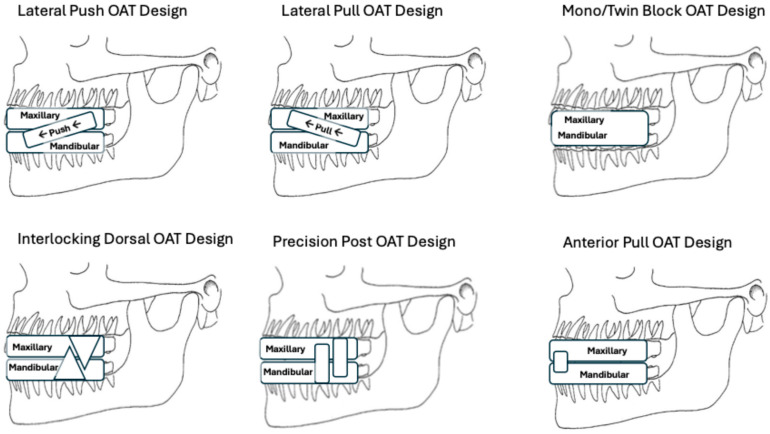
Line drawings of OAT design categories.

**Figure 2 bioengineering-12-00210-f002:**
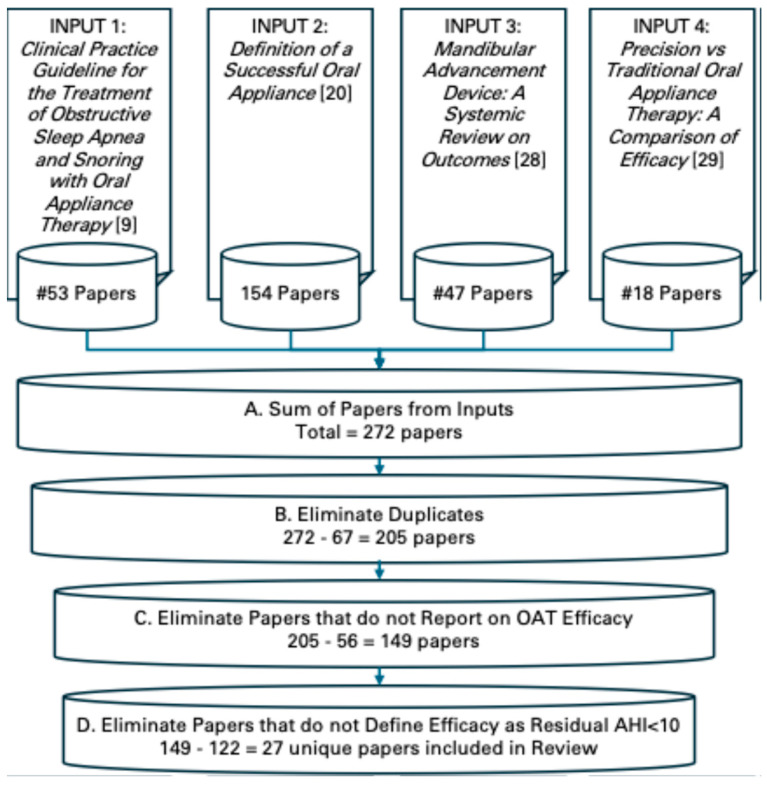
Search strategy flowchart.

**Figure 3 bioengineering-12-00210-f003:**
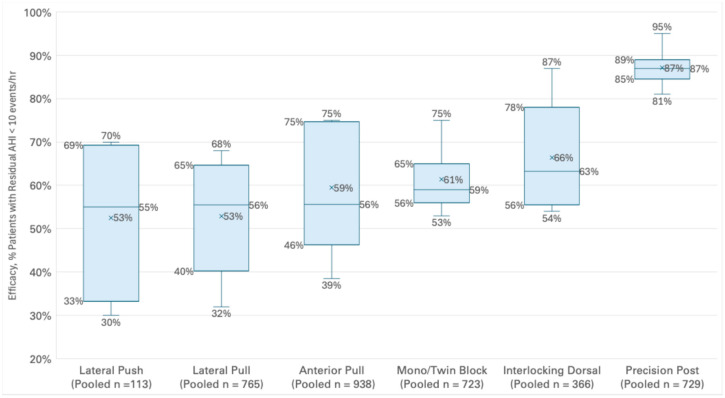
Box plot of efficacy by OA design category.

**Table 1 bioengineering-12-00210-t001:** Search results.

Category	Author and Reference	Sample	AHI < 10	DOI
Lateral Push	Randerath, 2022 [34]	20	30%	DOI: 10.1378/chest.122.2.569
Lateral Push	Ghazal, 2008 [35]	47	43%	DOI: 10.1111/j.1365-2869.2009.00738.x
Lateral Push	Gagnadoux, 2009 [36]	30	70%	DOI: 10.1183/09031936.00148208
Lateral Push	Bloch, 2000 [37]	16	67%	DOI: 10.1164/ajrccm.162.1.9908112
Lateral Pull	Vecchierini, 2016 [38]	369	64%	DOI: 10.1016/j.sleep.2015.05.020
Lateral Pull	Tegelberg, 2020 [39]	146	68%	DOI: 10.1080/00016357.2020.1730436
Lateral Pull	Kuna, 2005 [40]	21	43%	DOI: 10.1016/j.tripleo.2005.08.033
Lateral Pull	Isacsson, 2019 [41]	146	50%	DOI: 10.1093/ejo/cjy030
Lateral Pull	Isacsson, 2017 [42]	55	56%	DOI: 10.1007/s11325-016-1377-1
Lateral Pull	Henke, 1999 [43]	28	32%	DOI: 10.1164/ajrccm.161.2.9903079
Interlocking Flange	Vanderveken, 2024 [22]	91	56%	DOI: 10.1016/j.sleep.2024.02.035
Interlocking Flange	Van Haesendonck, 2016 [44]	112	57%	https://pubmed.ncbi.nlm.nih.gov/27097387/
Interlocking Flange	Schneiderman, 2021 [45]	36	69%	DOI: 10.1177/0022034520956977
Interlocking Flange	Remmers, 2013 [46]	67	87%	DOI: 10.5665/sleep.3048
Interlocking Flange	Mehta, 2001 [47]	24	54%	DOI: 10.1164/ajrccm.163.6.2004213
Interlocking Flange	de Ruiter, 2020 [48]	36	75%	DOI: 10.1007/s11325-020-02045-w
Dual Post	Stern, 2021 [49]	28	89%	DOI: 10.7759/cureus.15391
Dual Post	Silva, 2023 [50]	22	95%	DOI:10.1016/j.sleep.2023.11.228
Dual Post	Sall, 2023 [51]	91	89%	DOI: 10.7759/cureus.50107
Dual Post	Sall, 2021 [52]	115	87%	DOI: 10.1093/sleep/zsab072.433
Dual Post	Remmers, 2017 [53]	53	86%	DOI: 10.5664/jcsm.6656
Dual Post	Murphy, 2021 [54]	50	83%	DOI: 10.15331/jdsm.7208
Dual Post	Mosca, 2022 [55]	58	81%	DOI: 10.5664/jcsm.9758
Dual Post	Kang, 2024 [56]	24	88%	DOI: 10.1093/milmed/usac248
Dual Post	Dekow, 2023 [57]	288	86%	DOI: 10.1093/milmed/usab316
Block	Tegelberg, 2020 [39]	156	65%	DOI: 10.1080/00016357.2020.1730436
Block	Lettieri, 2011 [58]	203	64%	DOI: 10.5664/JCSM.1300
Block	Isacsson, 2019 [41]	156	53%	DOI: 10.1093/ejo/cjy030
Block	Isacsson, 2017 [42]	110	61%	DOI: 10.1007/s11325-016-1377-1
Block	Dutta, 2022 [59]	62	59%	DOI: 10.5664/jcsm.9742
Block	de Britto, 2013 [60]	19	58%	DOI: 10.1186/2196-1042-14-10
Block	Bloch, 2000 [37]	17	75%	DOI: 10.1164/ajrccm.162.1.9908112
Anterior Pull	Vanderveken, 2024 [22]	118	75%	DOI: 10.1016/j.sleep.2024.02.035
Anterior Pull	Schneiderman, 2021 [45]	36	56%	DOI: 10.1177/0022034520956977
Anterior Pull	Pancer, 1999 [61]	134	51%	DOI: 10.1378/chest.116.6.1511
Anterior Pull	Lettieri, 2011 [58]	602	74%	DOI: 10.5664/JCSM.1300
Anterior Pull	Ghazal, 2008 [35]	48	39%	DOI: 10.1111/j.1365-2869.2009.00738.x

**Table 2 bioengineering-12-00210-t002:** Two-sided Fisher’s exact tests.

	Lateral Push	Lateral Pull	Interlocking Dorsal	Precision Post	Mono/Twin Block
Lateral Pull	*p* = 0.103169				
Interlocking Dorsal	*p* = 0.010755	*p* = 0.081671			
Precision Post	*p* < 0.000001 *	*p* < 0.000001 *	*p* < 0.000001 *		
Mono/Twin Block	*p* = 0.067267	*p* = 0.763412	*p* = 0.137798	*p* < 0.000001 *	
Anterior Pull	*p* = 0.000441 *	*p* = 0.000119 *	*p* = 0.236170	*p* < 0.000001 *	*p* = 0.000401 *

Significance threshold adjusted for multiple comparisons (*p* < 0.00333). * denotes statistical significance.

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
