# Peer review of "Different Oral Appliance Designs Demonstrate Different Rates of Efficacy for the Treatment of Obstructive Sleep Apnea: A Review Article"

_bioengineering, 2025, doi:10.3390/bioengineering12020210_

Round 1

Reviewer 1 Report

Comments and Suggestions for Authors

This review article focuses on investigating the efficacy of oral appliance therapy devices for obstructive sleep apnea. The presentation of the article is very poor. The following comments need to be considered to improve the article.

1. Abbreviations of some keywords are already mentioned in the first paragraph of the introduction. However, the same is repeated in the third paragraph. Please maintain uniformity. 

2. Provide the images of the different OAT designs that are considered in the article. It will be very difficult for the reader to visualize without images.

3. Did the authors consider the reported OAT designs that are not mentioned in [9], [20], [24], [25]? Except [25], others are old papers. 

4. In what way this review article is different from the review works reported in [9], [20], [24] & [25]?

5. Why AHI of less than 10 events per hour is considered? Please justify.

6. Also discuss the disadvantages of the different OAT designs. Why the efficacy is less when compared to other designs?

7. Conclusion needs to be improved.

Reviewer 2 Report

Comments and Suggestions for Authors

Obstructive sleep apnea (OSA) is a sleep disorder characterized by the partial or complete collapse of the upper airway during sleep, resulting in reduced or absent breathing. Several medical devices have been developed to aid in the treatment of OSA, but their efficacy has not been systematically reviewed. This manuscript, titled “Different Oral Appliance Therapy Device Designs Are Associated with Significantly Different Efficacy for the Treatment of Obstructive Sleep Apnea: A Review Article,” evaluates whether variations in Oral Appliance Therapy (OAT) designs influence therapeutic outcomes. Analyzing data from 39 research studies, the authors conclude that OAT design significantly impacts treatment efficacy, with precision post designs showing the highest effectiveness.

While the review presents valuable insights, particularly regarding OAT selection for OSA treatment with potential clinical implications, several critical issues must be addressed before publication:

T   The authors found that precision post designs demonstrate the highest efficacy, but they fail to discuss the possible mechanisms behind this outcome. Has this been supported by other studies? If not, what factors might contribute to the observed differences? A deeper exploration of these aspects is necessary to enhance the scientific merit of the paper.

2.       The manuscript does not provide a clear explanation of how the included studies were selected. A flowchart illustrating the search strategy and criteria for selecting the 39 papers would improve transparency and reproducibility.

3.       The references for the 39 studies analyzed are not adequately documented. The summary in Table 1 lacks sufficient information for readers to verify and trace the data sources. A comprehensive list of these references must be included to ensure academic rigor and allow readers to evaluate the data independently.

      In addition, the title is quite wordy, a simpler title would help to clear the overall aims of this research. 

Round 2

Reviewer 2 Report

Comments and Suggestions for Authors

I'm happy to see that most of my concerns and comments are addressed by the authors, and have no further questions for its publication.